# Mesh Splatting for End-to-end Multiview Surface Reconstruction

**Ruiqi Zhang, Jiacheng Wu, and Jie Chen**
Department of Computer Science
Hong Kong Baptist University
`{csrqzhang, csjcwu, chenjie}@comp.hkbu.edu.h`

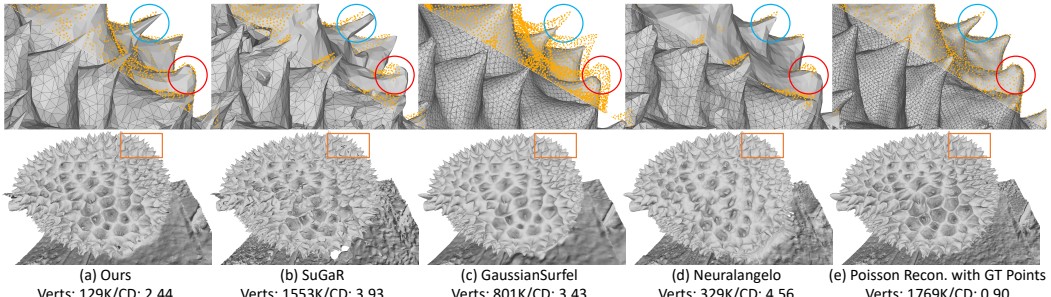

| (a) Ours | (b) SuGaR | (c) GaussianSurfel | (d) Neuralangelo | (e) Poisson Recon. with GT Points |
|---|---|---|---|---|
| Verts: 129K/CD: 2.44 | Verts: 1553K/CD: 3.93 | Verts: 801K/CD: 3.43 | Verts: 329K/CD: 4.56 | Verts: 1769K/CD: 0.90 |

Figure 1: **Comparison of reconstruction paradigms.** Yellow points denote ground-truth point clouds. "Verts" and "CD" denote the number of vertices and the Chamfer distance, respectively. (a) Our method optimizes meshes end-to-end and uses remeshing for topology control, achieving accurate surfaces with the fewest vertices. (b) SuGaR Guédon & Lepetit (2023) also optimizes meshes but relies on a single-layer Gaussian-splatting proxy and cannot perform remeshing, which limits accuracy. (c–d) As volumetric methods, GaussianSurfel Dai et al. (2024) and Neuralangelo Li et al. (2023) require a meshing step to extract surfaces, which accumulates errors and often yields unnecessarily dense meshes; note the misalignment between their meshes and the point clouds (red circle). (e) Poisson reconstruction on the *ground-truth points* shows that even with accurate point clouds, meshing can still introduce errors—e.g., omission of points (blue circle)—which constrains the practical upper bound of volumetric pipelines.

## Abstract

Surfaces are typically represented as meshes, which can be extracted from volumetric fields via meshing or optimized directly as surface parameterizations. Volumetric representations occupy 3D space and have a large effective receptive field along rays, enabling stable and efficient optimization via volumetric rendering; however, subsequent meshing often produces overly dense meshes and introduces accumulated errors. In contrast, pure surface methods avoid meshing but capture only boundary geometry with a single-layer receptive field, making it difficult to learn intricate geometric details and increasing reliance on priors (e.g., shading or normals). We bridge this gap by differentiably turning a surface representation into a volumetric one, enabling end-to-end surface reconstruction via volumetric rendering to model complex geometries. Specifically, we soften a mesh into multiple semi-transparent layers that remain differentiable with respect to the base mesh, endowing it with a controllable 3D receptive field. Combined with a splatting-based renderer and a topology-control strategy, our method can be optimized in about 20 minutes to achieve accurate surface reconstruction while substantially improving mesh quality.

## 1 Introduction

Surface reconstruction from images is a critical process for efficiently generating 3D assets across industries, including film production, video game development, and virtual/augmented reality. Among the diverse 3D representations (e.g., point clouds, signed distance fields, and meshes), meshes are widely preferred in practice due to their ease of manipulation and versatility Blender (2024).

Meshes can be extracted from volumetric fields through meshing techniques or optimized directly as surfaces. Volumetric methods include implicit Mildenhall et al. (2020); Wang et al. (2021), explicit Kerbl et al. (2023); Huang et al. (2024); Dai et al. (2024), and hybrid approaches Gu et al. (2025); Müller et al. (2022); Li et al. (2023). They occupy 3D space and enjoy a large effective receptive field along rays, enabling stable and efficient optimization via volumetric rendering. After optimization, meshes are typically extracted from volumetric representations using algorithms such as Marching Cubes Lorensen & Cline (1998), Marching Tetrahedra Shen et al. (2021), Poisson Reconstruction Kazhdan & Hoppe (2013), and TSDF-based methods Zhou et al. (2018). To improve surface reconstruction performance, prior work has focused on enhancing the accuracy and smoothness of the underlying surfaces represented volumetrically Wang et al. (2021); Li et al. (2023); Dai et al. (2024); Huang et al. (2024); Fu et al. (2022). For example, NeuS Wang et al. (2021) introduces surface constraints by reparameterizing the NeRF Mildenhall et al. (2020) density field as a signed distance field to improve surface smoothness; Neuralangelo Li et al. (2023) leverages hash-encoding networks Müller et al. (2022) to capture intricate details; Geo-NeuS Fu et al. (2022) incorporates sparse Structure-from-Motion (SfM) points as additional supervision; and 2DGS Huang et al. (2024) and GOF Yu et al. (2024) incorporate accurate normal/depth rendering into Gaussian Splatting Kerbl et al. (2023). Despite these advances, volumetric methods inevitably rely on a meshing step that can produce overly dense meshes and accumulate errors. Although remeshing Hoppe et al. (1993) can improve mesh quality, it is typically not differentiable within the optimization process and introduces additional error accumulation.

A parallel line of work, end-to-end mesh optimization Munkberg et al. (2022); Yang et al. (2025); Nicolet et al. (2021), avoids reliance on meshing and can control mesh quality through remeshing Hoppe et al. (1993) during optimization. However, meshes capture only boundary geometry with a single-layer receptive field, which hinders learning of intricate geometric details and increases reliance on priors (e.g., shading or normals). Some methods reparameterize meshes using volumetric proxies—such as tetrahedral grids in NvdiffRec Munkberg et al. (2022), point clouds in IMLS-Splatting Yang et al. (2025), and tetrahedral spheres in TetSphere Guo et al. (2024)—to improve topology stability during optimization. Nonetheless, they still optimize a single-layer surface with prior-based supervision: the primitive projected to image space remains the mesh, which lacks a volumetric receptive field and makes geometric detail recovery difficult; moreover, shading is hard to estimate under complex materials, and normal/depth estimation introduces additional errors.

To address the shortcomings that volumetric methods rely on error-prone meshing while surface-based methods lack sufficient 3D context and have a single-layer receptive field, we bridge the gap by differentiably converting surfaces into a pseudo-volumetric representation. Specifically, we soften a mesh into several semi-transparent layers that remain differentiable with respect to the base mesh, render them to image space using a splatting-based renderer, and composite multiple projected layers per pixel via volumetric rendering. The semi-transparent layers are randomly sampled around the base mesh, thereby enlarging the 3D receptive field for learning intricate details. In Fig. 2, we present a comparison between regular meshes and the proposed soft mesh. For regular meshes in Fig. 2(a), when the mesh does not overlap the real surface, multi-view observations can only optimize the color at point $A$ and provide little spatial gradient to move the geometry toward the true surface. In contrast, in Fig. 2(b), softening the base mesh into multiple semi-transparent layers increases the receptive field and creates overlap with the real surface, with transparency computed differentiably from the signed distance to the base mesh. During multi-view optimization, points $A$ and $B$ are both observed; since $A$ lies near the true surface, it exhibits similar appearance across views and receives a higher blending weight in volumetric compositing. This, in turn, reduces its signed distance and pulls the base mesh toward point $A$, ultimately deforming the mesh to accurately recover the surface.

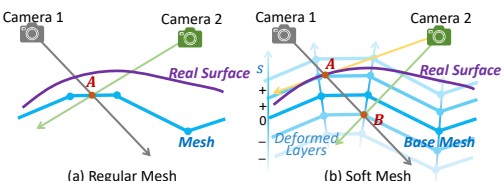

Figure 2: Comparison between regular meshes and soft mesh.

At the same time, the surface is represented by the base mesh, preserving the mesh's structural characteristics and enabling mesh quality control via remeshing. In practice, to maintain topology stability during mesh optimization, we build upon NvdiffRec Munkberg et al. (2022) by using a tetrahedral grid (DMTet) parameterization in the early optimization stage and employ Continuous Remeshing Palfinger (2022) for further topology control after convergence.

The contributions of this work are summarized as follows:

- We propose a method that bridges volumetric and surface representations by softening a mesh into several semi-transparent layers, thereby increasing the 3D receptive field for learning intricate details while preserving mesh characteristics, including topology control via remeshing techniques.
- We introduce a splatting-based renderer that efficiently renders the semi-transparent layers into images, enabling training in about 20 minutes.
- We present a hybrid topology-control strategy that combines DMTet meshing and Continuous Remeshing, improving topology stability during optimization and substantially enhancing final mesh quality.

## 2 RELATED WORKS

### 2.1 SURFACE RECONSTRUCTION WITH VOLUMETRIC REPRESENTATIONS

Volumetric representations occupy 3D space with nonzero density or transparency and are typically optimized via volumetric rendering Mildenhall et al. (2020). Representative examples include NeRF, which models scenes with a coordinate-based MLP that outputs density and radiance Mildenhall et al. (2020), and 3D Gaussian Splatting (3DGS), which parameterizes scenes with large sets of transparent ellipsoids Kerbl et al. (2023). Depending on the parameterization, approaches can be categorized as implicit Wang et al. (2021); Fu et al. (2022), explicit Huang et al. (2024); Dai et al. (2024); Chen et al. (2024), or hybrid Li et al. (2023); Gu et al. (2025). Early works such as NeRF Mildenhall et al. (2020) and 3DGS Kerbl et al. (2023) primarily target novel view synthesis, while recent efforts adapt these models to surface reconstruction with meshing techniques.

Mainstream research focuses on improving the accuracy and smoothness of the underlying surface modeled by volumetric representations. NeuS Wang et al. (2021) and VolSDF Yariv et al. (2021) introduce surface constraints by reparameterizing NeRF's density field as a signed distance field to promote smoothness. Neuralangelo Li et al. (2023) leverages multi-resolution hash encoding Müller et al. (2022) to capture fine geometric detail, while Geo-NeuS Fu et al. (2022) incorporates sparse Structure-from-Motion (SfM) points for additional supervision. In the explicit camp, GaussianSurfel Dai et al. (2024), 2DGS Huang et al. (2024), and GOF Yu et al. (2024) incorporate accurate normal and/or depth rendering into Gaussian splatting to better regularize surfaces.

After optimization, volumetric representations are converted to meshes via meshing techniques. Implicit models such as NeuS Wang et al. (2021) and Neuralangelo Li et al. (2023) typically evaluate the field on predefined grids and apply Marching Cubes Lorensen & Cline (1998). For explicit methods, GaussianSurfel Dai et al. (2024) applies Poisson Reconstruction Kazhdan & Hoppe (2013), while GOF Yu et al. (2024) employs Marching Tetrahedra Shen et al. (2021). To faithfully capture details, these pipelines often require high resolutions or depths, which can yield overly dense meshes, hindering practical deployment. Although downstream remeshing Hoppe et al. (1993) can improve quality, it is typically non-differentiable and can introduce additional error accumulation.

Our approach differs in that we optimize directly on meshes rather than on volumetric fields, avoiding meshing-induced error accumulation and low mesh quality. Moreover, we can control mesh topology during optimization via remeshing, yielding strong reconstruction accuracy even under tight vertex budgets suited to real applications.

### 2.2 SURFACE RECONSTRUCTION WITH MESHES

Surface representations concentrate mass on opaque surfaces, such as point clouds Yifan et al. (2019) and meshes Hoppe et al. (1993). We focus on meshes due to their manipulability and versatility. Meshes are typically rendered with differentiable rasterizers Ravi et al. (2020); Laine et al. (2020); Kato et al. (2020); Liu et al. (2019) and optimized with prior-based supervision, e.g., shading under known reflectance models or normals/depth predicted by monocular estimation networks Eftekhar et al. (2021). Several works improve topology stability during optimization: NvdiffRec Munkberg et al. (2022) and IMLS-Splatting Yang et al. (2025) reparameterize meshes as tetrahedral or cubic grids and rely on shading-based supervision, while SuGaR Guédon & Lepetit (2023) attaches flattened Gaussian ellipsoids to meshes and optimizes them jointly.

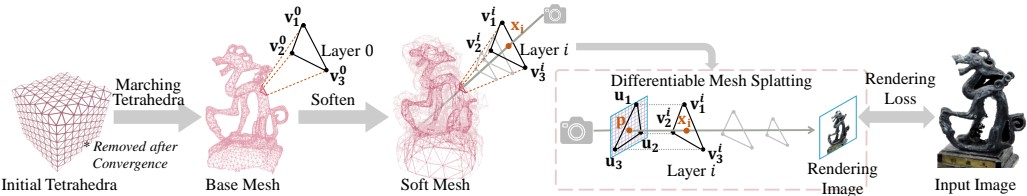

Figure 3: **Overview of the proposed method.** An initial tetrahedral grid stores signed-distance values at its vertices, and a base mesh is extracted using Marching Tetrahedra. The base mesh is then softened into multiple layers by offsetting vertices along their normals, transforming it from a surface into a pseudo-volumetric representation. The multi-layer mesh is rendered via the proposed Differentiable Mesh Splatting based on tile-based rasterization, and supervised by the input images through a rendering loss.

However, two limitations persist. First, the opaque, single-layer nature of meshes restricts their effective 3D receptive field, hindering the recovery of intricate geometry. Second, supervision via normals, depth, or shading is generally less accurate and informative than direct image supervision used in volumetric rendering. Normals from monocular predictors Eftekhar et al. (2021) can be noisy; depth captured by sensors Zhang (2012) carries measurement uncertainties; and real-world illumination often violates assumptions made by shading models Munkberg et al. (2022).

Our method addresses both issues. We soften the mesh into multiple semi-transparent layers sampled around the base surface, converting it into a pseudo-volumetric representation with a controllable 3D receptive field. This multi-layer structure enables optimization via volumetric rendering directly from image observations, reducing reliance on priors such as shading, normals, and depth.

### 2.3 OTHER MESH SOFTENING TECHNIQUES

We bridge surface and volumetric representations by softening meshes into multiple transparent layers. Related ideas have appeared primarily in novel view synthesis rather than surface reconstruction. For instance, Gaussian Shell Maps (GSM) Abdal et al. (2024) and DELIFFAS Kwon et al. (2024) build several layered shells around a base mesh (e.g., SMPL Loper et al. (2015)) to simulate light fields Levoy & Hanrahan (2023), enabling realistic rendering with coarse geometry. AdaptiveShell Wang et al. (2023) and Gaussian Frosting Guédon & Lepetit (2024) extract a base mesh from a pre-trained implicit model and envelop it with a two-layer shell, constraining radiance fields or Gaussian splats within the enclosed region for accelerated rendering. Quadrature Fields Sharma et al. (2024) and Volumetric Surfaces Esposito et al. (2024) construct grids or multi-layer structures around an extracted mesh to approximate SDF samples and achieve real-time rendering via rasterization-based intersection.

While these methods also place transparent layers around a base mesh, they typically target novel view synthesis and lack differentiability between the layers and the base mesh. Consequently, the base mesh remains fixed after initialization and cannot be refined using gradients from the semi-transparent layers. In contrast, our layers are differentiable with respect to the base mesh, allowing gradients from the volumetric rendering loss to update the base geometry, which is crucial for end-to-end surface reconstruction.

## 3 METHOD

An overview of the pipeline is shown in Fig. 3. We first soften the base mesh into several semi-transparent layers that remain differentiable with respect to the base mesh (Section 3.1). We then render these layers using Differentiable Mesh Splatting (Section 3.2). Following prior mesh-based methods Munkberg et al. (2022); Yang et al. (2025), we employ a hybrid topology-control strategy to maintain mesh quality during optimization (Section 3.3).

### 3.1 MESH SOFTENING

To construct the soft mesh in Fig. 2(b), we offset the vertices of the base mesh along their normals to form multiple sampled layers. We denote the base layer as $\mathcal{M}_0$ and the multi-layer mesh as

$\{\mathcal{M}_i\}_{i=1}^N$ derived from it. For the $j$-th base vertex $\mathbf{v}_j^0 \in \mathcal{M}_0$, the corresponding vertex on the $i$-th layer is calculated as:

$$\mathbf{v}_j^i = \mathbf{v}_j^0 + d_j^i \cdot \mathbf{n}_j, \tag{1}$$

where $d_j^i$ is the offset distance and $\mathbf{n}_j$ is the unit normal at $\mathbf{v}_j^0$.

We compute the transparency of each sampled layer from its signed distance to the base mesh. Since each softened vertex is generated by offsetting its source base vertex, it naturally has a closest-point correspondence. Let $\mathrm{stop}(\cdot)$ denote the stop-gradient operator. The signed distance at $\mathbf{v}_j^i$ is:

$$s_j^i = \mathrm{sign}(d_j^i) \left\| \mathrm{stop}(\mathbf{v}_j^i) - \mathbf{v}_j^0 \right\|_2, \tag{2}$$

and we stop gradients through $\mathbf{v}_j^i$ so that $s_j^i$ remains differentiable with respect to $\mathbf{v}_j^0$. Otherwise, substituting equation 1 into equation 2 collapses the expression to $s_j^i = \mathrm{sign}(d_j^i) \left\| d_j^i \cdot \mathbf{n}_j \right\|_2$, which is independent of $\mathbf{v}_j^0$ and cannot drive base-geometry updates.

We convert signed distances to alpha weights using a variant of the VolSDF mapping Yariv et al. (2021). Omitting indices for brevity, we define

$$\alpha = \begin{cases} \frac{1}{\beta}\left(1 - \frac{1}{2} e^{s/\beta}\right), & s < 0, \\ \frac{1}{2\beta} e^{-s/\beta}, & s \geq 0, \end{cases} \tag{3}$$

where $\beta > 0$ is a learnable parameter controlling how tightly the density concentrates around the base mesh (smaller $\beta$ yields sharper concentration).

In addition to the alpha $\alpha_j^i$, we attach per-vertex parameters needed for rendering: appearance features $\mathbf{f}_j^i$, vertex normals $\mathbf{n}_j^i$, viewing directions $\mathbf{r}_j^i$, and positions $\mathbf{x}_j^i$. The viewing direction $\mathbf{r}_j^i$ is the normalized vector from the camera center to the vertex and models view-dependent effects. These parameters are initialized on the base mesh and copied to different layers during rendering.

## 3.2 DIFFERENTIABLE MESH SPLATTING

Given the softened mesh, we render its semi-transparent layers into images with a splatting-based, differentiable pipeline that treats triangle faces as primitives. Using tile-based rasterization, we project triangle vertices to the image plane, find the triangles overlapping each pixel, and sort them from near to far by depth. For a triangle with vertices $\{\mathbf{v}_1^i, \mathbf{v}_2^i, \mathbf{v}_3^i\}$ covering pixel $\mathbf{p}$, we find the ray–triangle intersection $\mathbf{x_i}$ and compute its barycentric coordinates $\mathbf{w_i} = \{w_1, w_2, w_3\}$ with respect to the projected vertices $\{\mathbf{u}_1, \mathbf{u}_2, \mathbf{u}_3\}$, corrected by their depths $\{z_1, z_2, z_3\}$ as in Gu et al. (2025):

$$\mathbf{w_i} = \mathrm{correct}\left(\mathbf{p}, \{\mathbf{u}_1, \mathbf{u}_2, \mathbf{u}_3\}, \{z_1, z_2, z_3\}\right). \tag{4}$$

With $\mathbf{w_i}$, we obtain the per-intersection attributes $\{\alpha_\mathbf{i}, \mathbf{f_i}, \mathbf{n_i}, \mathbf{r_i}, \mathbf{x_i}\}$ by barycentric interpolation of the vertex-attached parameters. The color at the intersection is then predicted by an MLP:

$$\mathbf{c_i} = \mathrm{MLP}\left(\mathbf{f_i}, \mathbf{n_i}, \mathbf{r_i}, \mathrm{Hash}(\mathbf{x_i})\right), \tag{5}$$

where we use hash-encoded coordinate features $\mathrm{Hash}(\mathbf{x_i})$ Müller et al. (2022) to inject nonlinearity and avoid overly smooth interpolation within triangles.

Finally, we composite all overlapping triangles per pixel using the volumetric rendering equation:

$$\mathbf{C_p} = \sum_{i \in \mathcal{N}} \mathbf{c}_i \, \alpha_i \prod_{k=1}^{i-1} \left(1 - \alpha_k\right), \tag{6}$$

where $\mathcal{N}$ is the set of overlapping triangles sorted from near to far. The rendered image is supervised by the ground-truth image via a photometric loss, and gradients backpropagate through the layers to update the base mesh and its attached parameters.

### 3.3 Hybrid Topology Control

Direct mesh optimization is prone to defects that are difficult to repair. Following Nvd-iffRec Munkberg et al. (2022), we reparameterize the surface via Deep Marching Tetrahedra (DMTet) Shen et al. (2021) during the early stage to stabilize topology. Concretely, we initialize a tetrahedral grid with signed-distance values and extract the base mesh using DMTet. For a grid vertex at coordinate $\mathbf{x}_g$, we initialize the SDF as $\|\mathbf{x}_g - \mathbf{0}\|_2 - r$, yielding an initial spherical surface of radius $r$ centered at $\mathbf{0}$. Appearance-related parameters are stored on tetrahedral vertices and interpolated to mesh vertices.

This tetrahedral reparameterization provides robustness to topological artifacts early in training. Note that the SDF on tetrahedral vertices is unrelated to the signed-distance computation for softened mesh vertices in equation 2: the former stabilizes early geometry extraction, while the latter generates optimization gradients for the base mesh.

Because DMTet's resolution scales cubically and conflicts with explicit topology control (e.g., isotropic remeshing Hoppe et al. (1993)), we adopt the following strategy: after the DMTet stage converges, we freeze the mesh extracted from DMTet as a base mesh and disable further DMTet reparameterization. We then switch to Continuous Remeshing Palfinger (2022) to explicitly adjust topology and improve element quality, applying remeshing after each optimization step to maintain near-isotropic triangles and reduce defects.

### 3.4 Implementations

**Supervision.** Our approach inherits benefits from both surface and volumetric formulations. In addition to the image loss on volumetrically rendered images, we rasterize the base mesh Laine et al. (2020) and apply shading supervision following IMLS-Splatting Yang et al. (2025) and monocular normal supervision following GaussianSurfel Dai et al. (2024). We also include a mesh smoothness loss from PyTorch3D Ravi et al. (2020).

**Hyperparameters.** We initialize DMTet at resolution 128 within a $2.5\,\mathrm{m}$ bounding box (object-centric). We train with DMTet for 5,000 iterations, then remove it and continue for 10,000 iterations with Continuous Remeshing; the minimum edge length is controlled to be approximately $5\,\mathrm{mm}$. The base mesh is softened into 5 layers with offsets limited to $\pm 10\,\mathrm{cm}$. Training is performed on a single NVIDIA V100 ($32\,\mathrm{GB}$) GPU and takes about 20 minutes per scene.

## 4 Experiments

### 4.1 Experimental Setup

The main contribution of our method is to soften a mesh, bridging the gap between surface and volumetric representations. This increases the effective 3D receptive field for detailed geometry recovery while maintaining the topology-control capabilities of meshes, enabling accurate, high-quality reconstructions. To showcase reconstructed detail, we evaluate on object-centric datasets and report Chamfer Distance (in *cm*) and vertex counts: DTU Jensen et al. (2014), which features complex illumination and relatively simple geometry, and BlendedMVS Yao et al. (2020), which contains more intricate shapes. Although our approach is not theoretically confined to object-centric settings, it can be applied to scene-level datasets with modest modifications to the mesh reparameterization (as in IMLS-Splatting Yang et al. (2025)). To validate this, we initialize scene-level experiments with coarse meshes produced by GaussianSurfel Dai et al. (2024) and further optimize them using our method, yielding realistic reconstructions, as shown in Fig. 8.

For a comprehensive comparison, we include representative methods from multiple paradigms: implicit (NeuS Wang et al. (2021)), hybrid (Neuralangelo Li et al. (2023)), explicit Gaussian-based (GOF Yu et al. (2024), GaussianSurfel Dai et al. (2024), 2DGS Huang et al. (2024)), and mesh-based (SuGaR Guédon & Lepetit (2023), IMLS-Splatting Yang et al. (2025)).

### 4.2 Comparison with SOTA Methods on Object-centric Datasets

In Fig. 1, we illustrate meshing-induced error accumulation and unnecessarily dense meshes in volumetric pipelines, as well as the limitations of prior mesh-based methods. The quantitative com-

Table 1: Surface reconstruction accuracy on DTU Jensen et al. (2014) dataset. Best results are highlighted as 1st and 2nd. Approximate vertex counts (in thousands) and training time (minutes) are shown on the right.

| Method | 24 | 37 | 40 | 55 | 63 | 65 | 69 | 83 | 97 | 105 | 106 | 110 | 114 | 118 | 122 | Mean | Verts | Training |
|---|---|---|---|---|---|---|---|---|---|---|---|---|---|---|---|---|---|---|
| NeuS | 0.83 | 0.98 | 0.56 | 0.37 | 1.13 | 0.59 | 0.60 | 1.45 | 0.95 | 0.78 | 0.52 | 1.43 | 0.36 | 0.45 | 0.45 | 0.76 | 1000 | 600 |
| NeuS2 | 0.56 | 0.76 | 0.49 | 0.37 | 0.92 | 0.71 | 0.76 | 1.22 | 1.08 | 0.63 | 0.59 | 0.89 | 0.40 | 0.48 | 0.55 | 0.70 | 1000 | 3 |
| Neuralangelo | 0.45 | 0.74 | 0.33 | 0.34 | 1.05 | 0.54 | 0.53 | 1.33 | 1.05 | 0.72 | 0.43 | 0.69 | 0.34 | 0.38 | 0.42 | 0.62 | 1000 | 600 |
| Surfel | 0.66 | 1.07 | 0.58 | 0.49 | 0.93 | 1.08 | 0.87 | 1.29 | 1.53 | 0.76 | 0.86 | 1.87 | 0.53 | 0.67 | 0.60 | 0.92 | 1000 | 6 |
| 2DGS | 0.49 | 0.79 | 0.37 | 0.43 | 0.94 | 0.92 | 0.83 | 1.24 | 1.25 | 0.65 | 0.64 | 1.58 | 0.43 | 0.69 | 0.50 | 0.78 | 300 | 9 |
| GOF | 0.50 | 0.82 | 0.37 | 0.37 | 1.12 | 0.74 | 0.73 | 1.18 | 1.29 | 0.68 | 0.77 | 0.90 | 0.42 | 0.66 | 0.49 | 0.74 | 1000 | 18 |
| SuGaR | 1.47 | 1.33 | 1.13 | 0.61 | 2.25 | 1.71 | 1.15 | 1.63 | 1.62 | 1.07 | 0.79 | 2.45 | 0.98 | 0.88 | 0.79 | 1.33 | 1000 | 52 |
| IMLS-Splatting | 0.52 | 1.02 | 0.37 | 0.32 | 0.86 | 0.50 | 0.48 | 1.15 | 0.76 | 0.59 | 0.37 | 0.67 | 0.33 | 0.33 | 0.34 | 0.57 | 300 | 11 |
| Ours w/o MS | 0.49 | 1.00 | 0.86 | 0.40 | 0.85 | 0.99 | 0.63 | 1.23 | 1.24 | 0.62 | 0.67 | 0.63 | 0.35 | 0.51 | 0.49 | 0.73 | 300 | 20 |
| Ours | 0.46 | 0.73 | 0.49 | 0.43 | 0.77 | 0.82 | 0.65 | 1.03 | 0.95 | 0.52 | 0.58 | 0.59 | 0.37 | 0.44 | 0.42 | 0.62 | 300 | 23 |

Table 2: Surface reconstruction accuracy on BlendedMVS Yao et al. (2020) dataset. Best results are highlighted as 1st and 2nd.

| Method | Basketball | Bear | Bread | Camera | Clock | Cow | Dog | Doll | Dragon | Durian | Fountain | Gundam | House | Jade | Man | Monster | Sculpture | Stone | Mean |
|---|---|---|---|---|---|---|---|---|---|---|---|---|---|---|---|---|---|---|---|
| NeuS | 2.96 | 3.00 | 2.85 | 2.61 | 2.75 | 2.04 | 2.75 | 2.17 | 2.95 | 3.14 | 3.03 | 1.62 | 3.23 | 4.25 | 2.29 | 1.92 | 2.10 | 2.51 | 2.68 |
| Surfels | 1.59 | 1.52 | 1.41 | 1.75 | 5.08 | 3.21 | 2.97 | 2.32 | 2.93 | 4.01 | 2.65 | 0.97 | 1.76 | 3.49 | 2.23 | 1.36 | 3.07 | 2.02 | 2.46 |
| SuGar | 8.00 | 9.73 | 7.65 | 7.77 | 9.21 | 8.69 | 9.27 | 8.75 | 9.76 | 8.04 | 9.05 | 7.32 | 7.28 | 10.7 | 9.29 | 8.20 | 8.98 | 9.19 | 8.71 |
| IMLS-Splatting | 2.48 | 1.86 | 2.69 | 3.61 | 2.96 | 2.80 | 2.85 | 2.32 | 2.39 | 3.35 | 2.83 | 1.78 | 3.01 | 5.10 | 2.61 | 1.99 | 2.04 | 2.85 | 2.75 |
| Ours w/o MS | 1.41 | 1.34 | 0.84 | 2.02 | 2.38 | 1.52 | 2.02 | 2.58 | 1.74 | 2.56 | 2.64 | 0.98 | 1.81 | 3.86 | 1.73 | 1.63 | 1.98 | 1.92 | 1.94 |
| Ours | 1.26 | 1.16 | 0.75 | 1.75 | 1.99 | 1.30 | 2.08 | 2.02 | 1.57 | 2.36 | 2.37 | 0.90 | 1.94 | 3.27 | 1.52 | 1.30 | 1.64 | 1.64 | 1.71 |

parisons in Tables 1 and 2 further show that our method reaches state-of-the-art (SOTA) accuracy with fewer vertices, attributable to explicit control over mesh quality and topology. We note a slight accuracy gap versus IMLS-Splatting Yang et al. (2025) on DTU (Table 1). IMLS-Splatting converts point clouds to a grid, extracts meshes, and optimizes them with a shading loss. To isolate the effect of our softening-based volumetric supervision from 3D representation choices, we ablate the softening stage and train with shading-only supervision ("Ours w/o MS" in Tables 1 and 2). This ablation underperforms the full model, indicating that our end-to-end volumetric optimization provides stronger geometric supervision than shading alone. We expect further gains if our softening mechanism is paired with more flexible 3D parameterizations like those used in IMLS-Splatting.

Visual comparisons highlight the qualitative advantages of our approach. In Fig. 4, our method recovers clean, smooth surfaces and fine structures, such as the window frames in *Scan24*, scissor blades in *Scan37*, the eyes in *Scan106*, and facial lines in *Scan114*. By contrast, GaussianSurfel Dai et al. (2024) and GOF Yu et al. (2024) represent scenes with discrete Gaussian primitives that lack the inherent surface smoothness of meshes, leading to broken structures in *Scan37* and surface artifacts in *Scan114*, as well as missing details like the eyes in *Scan106* and smile lines in *Scan114*. These failures are consistent with error accumulation introduced by downstream meshing. On BlendedMVS (Fig. 5), broken bases (e.g., *Dragon*) and missing indentations (e.g., *Stone*) similarly reflect the lack of surface smoothness and meshing-induced artifacts.

Compared to IMLS-Splatting Yang et al. (2025) in Fig. 5, we observe the limitations of shading-only supervision, such as noisy surfaces on *Monster* and *Stone*. Moreover, the grid resolution in IMLS-Splatting scales cubically, whereas our pipeline optimizes a pure mesh in later stages, enabling higher vertex densities to capture more detail. To further illustrate the limitations of shading-only training, we show "Ours w/o MS" in Fig. 5: although this variant uses denser meshes, it still fails to reproduce the indentation at the top of *Stone*. This confirms that without the multi-layer softening (and its enlarged 3D receptive field), shading-only supervision is insufficient for reliably recovering intricate geometry.

## 4.3 RESULTS ON NeRF SYNTHETIC DATASET

We provide results on the NeRF Synthetic Dataset Martin-Brualla et al. (2021) to highlight our method's effectiveness in capturing thin structures. As shown in Fig. 6, we compare our approach with Gaussian Surfel Dai et al. (2024). The results demonstrate that our method can accurately represent thin structures and consistently outperforms Gaussian Surfel, particularly for the mast of the *ship* and the vase in the *ficus* example. Furthermore, our approach achieves these improvements with significantly reduced vertex counts and optimized mesh topology, as evident in the leaf blade of the *ficus*.

These improvements can be attributed to two main reasons. First, our method introduces accurate volumetric supervision to meshes, enabling the model to learn flexible geometric details, such as

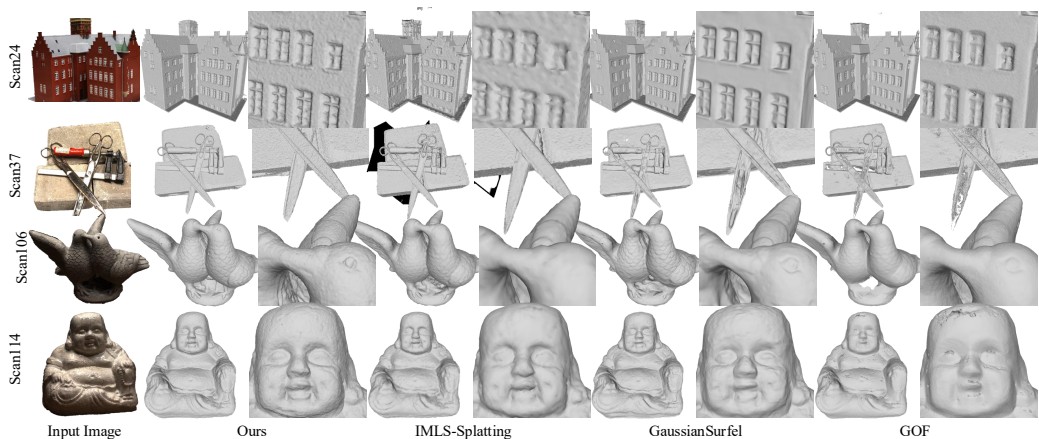

Figure 4: **Qualitative comparison on DTU dataset.**

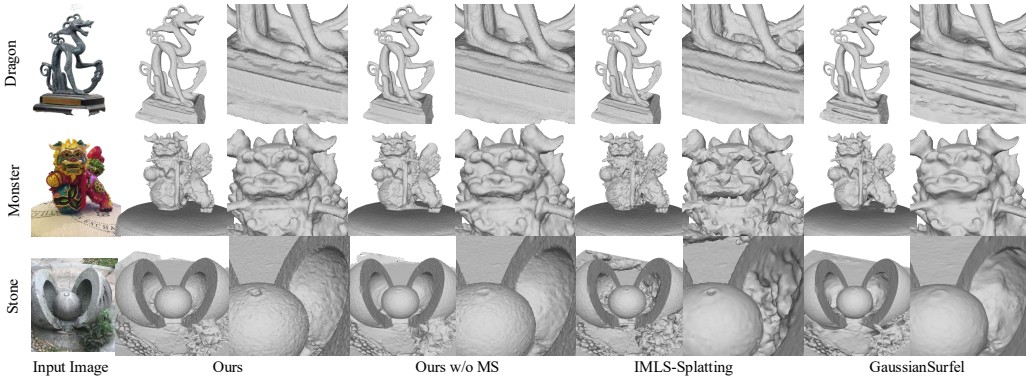

Figure 5: **Qualitative comparison on BMVS dataset.**

the vase's wrinkles and the ship's mast. Second, by employing remeshing during optimization and circumventing post-optimization meshing, our method achieves better mesh topology with fewer vertices, which is vital for downstream applications such as physical simulation.

We also observe that the cable of the *ship* is not produced by either method. For our method, this indicates that isotropic remeshing may not be well-suited for extremely thin structures, such as cables or other fine structures like human hair. In this case, adaptive remeshing could be explored to generate slender triangular faces for cable-like structures while maintaining isotropic faces for flat surfaces. For Gaussian Surfel, although Gaussian ellipsoids can be positioned to represent cables, it fails to reconstruct meshes from them due to limited meshing resolution, resulting in accumulated errors from ellipsoids-to-mesh conversion.

## 4.4 ABLATION

### 4.4.1 RENDERING EFFICIENCY: MESH SPLATTING VS. ITERATIVE MESH RASTERIZATION

To assess the efficiency of our splatting-based renderer, we implement an alternative based on iterative mesh rasterization, using depth peeling in Nvdiffrast Laine et al. (2020), to render the softened mesh (i.e., multiple semi-transparent layers) into images. Table 3 reports results on DTU *scan122* and includes Gaussian Splatting (as implemented in GaussianSurfel Dai et al. (2024)) as a reference. The original image resolution is 1600×1200; we evaluate at 1/4, 1/2, and full resolution, and report GPU memory usage and training time for each method.

Table 3: Memory and training time for Mesh Splatting (MS), Iterative Mesh Rasterization (IMR), and Gaussian Splatting (GS) on DTU *scan122* at different image scales.

| | Memory (GB) ↓ | | | Training (Minutes) ↓ | | |
|---|---|---|---|---|---|---|
| Resize Scale | 1/4 | 1/2 | 1 | 1/4 | 1/2 | 1 |
| GS | 1 | 2 | 6 | 3 | 5 | 13 |
| IMR | 8 | 25 | OOM | 40 | 90 | N/A |
| MS | 2 | 4 | 13 | 12 | 15 | 22 |

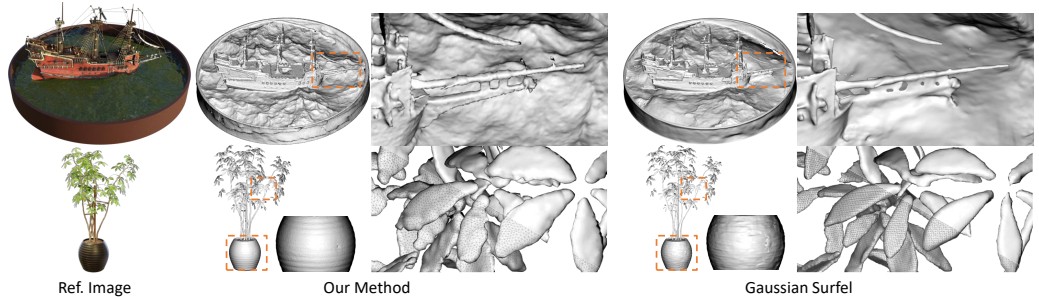

Figure 6: **Reference results on the NeRF Synthetic Dataset Martin-Brualla et al. (2021).** We present results for the *ship* and *ficus* examples to better visualize thin structures.

The results indicate that Mesh Splatting (MS) is significantly more efficient than iterative mesh rasterization (IMR). At 1/4 resolution, MS consumes 2 GB of memory compared to 8 GB for IMR, and on an NVIDIA V100 (32 GB) GPU, IMR runs out of memory (OOM) at full resolution, whereas MS remains feasible. Gaussian Splatting (GS) is still more efficient than MS, suggesting room for engineering improvements to the MS implementation, such as culling invisible triangles and adaptively controlling mesh vertex density.

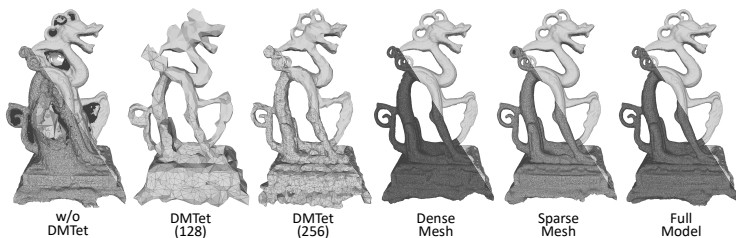

Figure 7: **Qualitative ablations.** Hybrid topology control (DMTet + Continuous Remeshing) captures global topology and fine details more reliably than using either component alone.

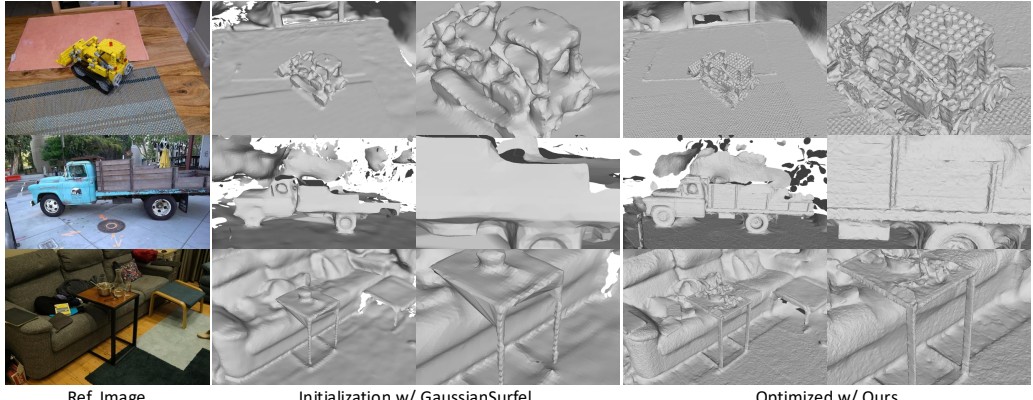

Figure 8: **Reference results on scene-level datasets.** Initial coarse meshes from GaussianSurfel Dai et al. (2024) are refined with our softening and splatting, plus Continuous Remeshing. The data is from Mip-NeRF360 Barron et al. (2022) and TandT Knapitsch et al. (2017).

### 4.4.2 EFFECT OF HYBRID TOPOLOGY CONTROL

We combine DMTet Shen et al. (2021) (early stage) with Continuous Remeshing Palfinger (2022) (later stage) to jointly stabilize global topology and improve element quality. As shown in Table 4 and Fig. 7, removing DMTet ("w/o DMTet") fails to capture global topology (e.g., holes), whereas using DMTet alone at resolution 128 ("DMTet (128)") yields sparse meshes and inferior accuracy. Increasing to resolution 256 improves accuracy but still lacks fine detail, and its training time is already comparable to the full model due to cubic scaling. In contrast, the hybrid strategy captures global topology with sparse DMTet and recovers fine detail with explicit remeshing, providing the best overall accuracy and mesh quality.

Table 4: Ablation metrics.

|  | w/o DMTet | DMTet (128) | DMTet (256) | Dense Mesh | Sparse Mesh | Full Model |
|---|---|---|---|---|---|---|
| Memory(GB) | 7 | 6 | 8 | 28 | 15 | 23 |
| Training | 18 | 15 | 23 | 35 | 19 | 25 |
| Vertices | 80K | 2K | 10K | 487K | 127K | 306K |
| CD | 3.79 | 6.94 | 4.20 | 1.67 | 1.66 | 1.57 |

### 4.4.3 VERTEX NUMBER CONTROL

Continuous Remeshing exposes a minimum edge-length parameter that directly controls mesh density. As seen in Table 4 and Fig. 7, our method is robust to vertex count within a wide range. In practice, we set the minimum edge length to approximately 5 mm, yielding around 300k vertices per object and a favorable balance between accuracy and efficiency.

### 4.5 LIMITATIONS

While the proposed framework—mesh softening, splatting-based rendering, and hybrid topology control—is general, scalability to very large scenes can be constrained by tetrahedral-grid resolution and GPU memory. This limitation does not affect our central contribution: converting a mesh into a pseudo-volumetric multi-layer representation increases the effective 3D receptive field and enables image-supervised optimization while preserving mesh characteristics and topology control.

To illustrate applicability in scene-level settings, we initialize with coarse meshes obtained from GaussianSurfel and then optimize them using our softening and splatting modules, with Continuous Remeshing to maintain element quality. As shown in Fig. 8, our method consistently adds geometric detail over the initialization. A remaining failure case occurs when the base mesh is far from the true surface (e.g., distant background): softening into a thin band around the base surface may not sufficiently overlap the target region to provide volumetric gradients. Addressing this at scale will likely require modifying the mesh reparameterization and sampling strategy (e.g., adaptive layer bandwidths or hierarchical softening), which we leave for future work.

## 5 CONCLUSION

Mesh Splatting softens a base mesh into differentiable semi-transparent layers and renders them volumetrically, enabling end-to-end mesh optimization with a controllable 3D receptive field and direct image supervision. With an efficient splatting renderer and hybrid topology control (DMTet early, Continuous Remeshing later), we achieve strong accuracy with fewer vertices and shorter training time; ablations confirm the importance of multi-layer softening and the superiority of the hybrid control. Future work will scale to larger scenes and further optimize the renderer (e.g., triangle culling and adaptive density).

### ACKNOWLEDGMENTS

This research was supported by the Theme-based Research Scheme, Research Grants Council of Hong Kong (T45-205/21-N), and the Guangdong and Hong Kong Universities "1+1+1" Joint Research Collaboration Scheme (2025A0505000003).

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

## A PSEUDO CODE FOR TRAINING

To aid understanding of our method, we present pseudocode for Mesh Splatting Training.

---

**Algorithm 1:** Mesh Splatting Training

---

**Input:** Tetrahedral grid $\mathcal{G}$ with vertex SDF $\mathcal{S}$;
Training images $\{I_{\text{GT}}^v\}$ and cameras $\{C^v\}$;
Total iterations $T$; warmup $T_{\text{DMTet}}$; remesh period $T_{\text{remesh}}$;
Number of soften layers $N$; renderer $\mathcal{R}_{\text{splat}}$;
Photometric loss $\rho(\cdot, \cdot)$.
**Output:** Final base mesh $\mathcal{M}_0$
**for** $t = 1$ **to** $T$ **do**

    // Topology control schedule
    **if** $t \leq T_{\text{DMTet}}$ **then**
        // Early stage:  re-extract mesh from current SDF
        $\mathcal{M}_0 \leftarrow \text{DMTet}(\mathcal{G}, \mathcal{S})$
    **else**
        // Late stage:  refine mesh quality/topology
        $\mathcal{M}_0 \leftarrow \text{Remeshing}(\mathcal{M}_0)$
    **end**
    // Generate softened layers on-the-fly
    $\{\mathcal{M}_i\}_{i=1}^N \leftarrow \text{Soften}(\mathcal{M}_0, N)$
    // Render a view
    Select a view $v$
    $I_{\text{pred}}^v \leftarrow \mathcal{R}_{\text{splat}}(\{\mathcal{M}_i\}_{i=1}^N, C^v)$
    // Photometric loss
    $L \leftarrow \rho(I_{\text{pred}}^v, I_{\text{GT}}^v)$
    // Backpropagation and parameter updates
    **if** $t \leq T_{\text{DMTet}}$ **then**
        Update $\mathcal{S}$ using $\nabla_{\mathcal{S}} L$
    **else**
        Update $\mathcal{M}_0$ (vertex positions/attributes) using $\nabla_{\mathcal{M}_0} L$
    **end**
**end**

---

## B COMPARISON WITH NEURALANGELO

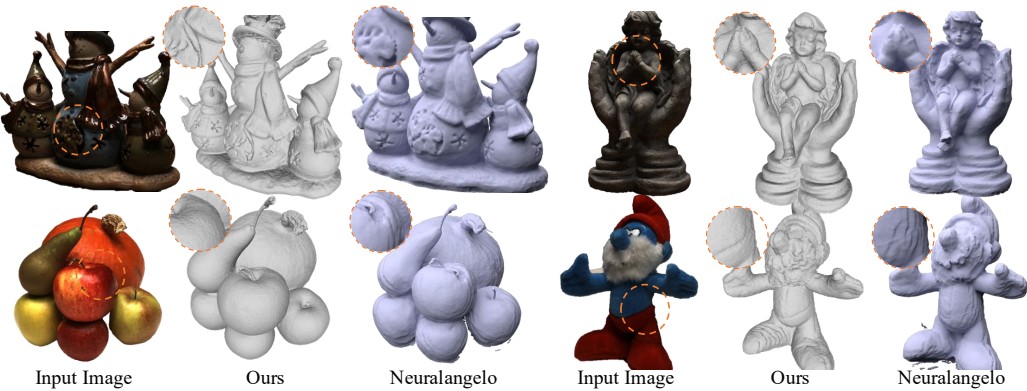

Input Image    Ours    Neuralangelo    Input Image    Ours    Neuralangelo

Figure 9: **Qualitative comparison on Neuralangelo.**

We additionally compare against Neuralangelo Li et al. (2023). Owing to its prohibitive training time, we rely on the Neuralangelo results reported by IMLS-Splatting Yang et al. (2025). As shown in Table 1, our method achieves comparable accuracy while using far fewer vertices and substantially shorter training time. Qualitatively (Fig. 9), our reconstructions capture details on par with

Neuralangelo (e.g., the hands), and exhibit smoother surfaces in reflective regions, as seen in the two leftmost examples.

## C    SHADING SUPERVISION

In addition to the image reconstruction loss computed on volumetrically rendered images, our method incorporates direct shading supervision on the rasterized base mesh, closely following IMLS-Splatting. Specifically, we utilize Differentiable Mesh Rasterization Laine et al. (2020) to project the base mesh onto the image plane, producing a foreground mask $\mathbf{I_m} \in R^{H \times W}$, a normal map $\mathbf{I_n} \in R^{H \times W \times 3}$, and a 2D feature map $\mathbf{I_f} \in R^{H \times W \times D}$ (interpolated from vertex-attached features). Similar to IMLS-Splatting, we decode the feature map into spatially-varying surface properties at each pixel, including diffuse color $\mathbf{c_d}$, specular tint $\mathbf{s}$, and specular feature $\mathbf{f_s}$ via an MLP $\Phi$. Additionally, we employ a lightweight MLP $\Phi_s$ to predict the specular color for each pixel given its specular feature $f_s$, view directions $\omega$, and reflection direction $\omega_r$, enabling efficient modeling of complex occlusions and view-dependent effects. The final color $\mathbf{c}$ for each pixel is computed as:

$$\mathbf{c} = \mathbf{c_d} + \mathbf{s} \odot \Phi_s(\mathbf{f_s}, \omega, \omega_r), \tag{7}$$

where $\omega$ denotes the direction vector from the camera to the corresponding surface point on the mesh, and the reflect direction is defined as $\omega_{\mathbf{r}} = 2(-\omega \cdot \mathbf{n})\mathbf{n} + \omega$, with surface normal $\mathbf{n}$ derived from $\mathbf{I_n}$. The final rasterized color image $\mathbf{I_c}$ is obtained by applying the foreground mask $\mathbf{I_m}$ to the computed color $\mathbf{c}$ at each pixel.

## D    OTHER VISUALIZATION RESULTS

We provide the mesh files used in the teaser in the Supplementary Material: meshes generated by GaussianSurfel, Neuralangelo, and our method (SuGaR is omitted due to its large file size). We also include a mesh produced by Poisson reconstruction of the ground-truth point clouds, and a video showing the optimization process for the BlendedMVS Yao et al. (2020) *dragon*, which takes approximately 23 minutes.

## E    STATEMENT ON THE USE OF LARGE LANGUAGE MODELS

We would like to clarify that Large Language Models (LLMs) were exclusively utilized for linguistic refinement and the polishing of the manuscript. LLMs were not involved in the development of the research concepts, experimental design, analysis, or the formulation of the core ideas presented in this paper. All scientific contributions and methodological innovations originated solely from the authors.

