# OpenReview forum: "Mesh Splatting for End-to-end Multiview Surface Reconstruction"
_ICLR.cc/2026/Conference — ICLR 2026 Poster_

### Official Review · Reviewer_hBem · 2025-10-24

**Soundness:** 3
**Presentation:** 3
**Contribution:** 2
**Rating:** 6
**Confidence:** 4

**Summary:**

This paper proposes a novel framework that bridges the gap between volumetric and surface-based 3D reconstruction methods. The core idea is to soften a base mesh into multiple semi-transparent, differentiable layers, effectively transforming it into a pseudo-volumetric representation while preserving the explicit structure of the mesh. These softened layers are rendered through a splatting-based differentiable renderer and optimized end-to-end with image-level supervision. By combining volumetric gradient propagation with explicit topology control, the method achieves high-quality surface reconstructions with fewer vertices and significantly reduced training time compared to existing volumetric or mesh-based approaches.

**Strengths:**

1. The proposed concept of differentiably softening meshes into volumetric layers is both innovative and practical. It effectively bridges the gap between surface- and volume-based approaches, enabling volumetric supervision while preserving explicit mesh controllability and topology refinement.
2. The paper demonstrates convincing quantitative and qualitative results on the DTU and BlendedMVS benchmarks, showing clear visual and numerical improvements over prior methods.
3. The overall presentation is clear, well-structured, and easy to follow.

**Weaknesses:**

1. Since the proposed framework claims to reconstruct intricate geometric details, I recommend including comparisons on the Ship or Ficus scenes from the NeRF Synthetic dataset, as these cases involve thin structures that are particularly challenging to recover accurately.
2. The overall novelty of the framework appears slightly limited, as the pipeline is largely built upon DMTet and splatting. Prior approaches such as VMesh [1] and Radiance Surfaces [2] have also explored incorporating volumetric supervision into surface-based representations.
3. The method still relies on tetrahedral grids, which may restrict its scalability and generalization to large-scale or unbounded scenes, potentially limiting its broader applicability.

[1] Guo, Y. C., Cao, Y. P., Wang, C., He, Y., Shan, Y., & Zhang, S. H. (2023, December). Vmesh: Hybrid volume-mesh representation for efficient view synthesis. In SIGGRAPH Asia 2023 Conference Papers (pp. 1-11).

[2] Zhang, Z., Roussel, N., Muller, T., Zeltner, T., Nimier-David, M., Rousselle, F., & Jakob, W. (2025, August). Radiance surfaces: Optimizing surface representations with a 5D radiance field loss. In Proceedings of the Special Interest Group on Computer Graphics and Interactive Techniques Conference Conference Papers (pp. 1-10).

**Questions:**

1.  The paper primarily evaluates object-centric datasets, while scene-level reconstructions are only briefly discussed in the ablation section. For larger-scale environments (e.g., Mip-NeRF360), it would be helpful to clarify how the authors adapt or extend the DMTet initialization to achieve more stable and accurate reconstruction at scene scale.
2. Since the framework relies on DMTet for base mesh reconstruction, exploring FlexiCubes as an alternative could potentially enhance the quality and robustness of the initial mesh. Such a comparison or discussion would further strengthen the technical completeness of the work.

---

> ### Author Response · Authors · 2025-11-15
> **part 1**
>
> ## 1. Response for Limited Novelty compared with DMTet and Splatting
>
> The reviewer comments that **"The overall novelty of the framework appears slightly limited, as the pipeline is largely built upon DMTet and splatting."** We respectfully disagree. Our main novelty lies in introducing a fundamental optimization strategy for meshes by converting meshes into a pseudo-volumetric representation. This approach is significantly more efficient, accurate, and robust than traditional mesh supervision signals, as we further explain below. In our pipeline, DMTet is a technical choice to stabilize topology during early optimization, while triangle-based splatting accelerates rendering. These are supporting engineering decisions and do not diminish our central innovation, which is a fundamental and effective supervision mechanism for end-to-end mesh optimization. Furthermore, although prior work has explored ellipsoid-based and tetrahedra-based rendering [3, 4], we are the first to achieve triangle-based rendering with underlying volumetric composition. Thus, the triangle-based splatting itself also contributes additional novelty.
>
>
> **Limitations of existing mesh-based surface reconstruction**: Mesh-based surface reconstruction enables end-to-end optimization and avoids the post-optimization meshing typically required by volumetric methods for mesh extraction, thereby reducing accumulated errors and uncontrollable topology resulting from the conversion from volumetric representations to meshes. However, commonly used geometric supervision signals for meshes have notable drawbacks:
>
> - **Normal supervision** requires pretraining a normal estimator to predict normal vector images from color images, using the pixel-wise normal vectors to supervise the mesh normals. However, estimating normals from color images introduces unavoidable estimation errors and dataset bias, especially when input images do not align with the training data distribution.
> - **Shading supervision** relies on a predefined image formation (reflectance) model, such as the Phong model, which typically takes viewing direction, surface normals, light source directions, and surface material parameters as input. These inputs are aggregated into outgoing radiance and compared with the color image. However, a single reflectance model rarely fits all objects and materials, and jointly optimizing geometry, materials, and lighting usually introduces ambiguity.
> - **Depth supervision** uses depth captured by LiDAR or depth sensors, with pixel-wise depth values used to supervise the pixel-wise distance from the mesh to the camera. However, captured depth often contains nontrivial measurement noise.
>
>
> **Our core contribution**: We propose a fundamental optimization strategy for meshes—an efficient, robust, and accurate alternative to normal/shading/depth supervision. We convert a single mesh into a multi-layer mesh by deforming several layers from a base mesh at randomized offsets. This ensures that some layer is likely to intersect the true surface and can be used to perceive the real surface, providing spatial gradients that pull the base mesh toward the correct geometry. The key benefits are:
> - **Efficiency**: Tile-based rasterization for the multi-layer mesh enables fast optimization.
> - **Robustness**: Supervision comes directly from color images, without reliance on learned priors or fixed reflectance models.
> - **Accuracy**: The enlarged receptive field increases the likelihood of receiving informative geometry gradients.
>
>
> ## 2. Comparison with VMesh and Radiance Surfaces
>
> The reviewer comments that **"Prior approaches such as VMesh [1] and Radiance Surfaces [2] have also explored incorporating volumetric supervision into surface-based representations."** However, these methods differ from ours in that they incorporate volumetric supervision to optimize **volumetric fields (SDF/occupancy)** rather than **meshes** directly. As a result, they still require meshing algorithms such as Marching Cubes or TSDF fusion as post-processing, which introduces accumulated errors during the conversion from fields to meshes and can result in uncontrollable topology. Specifically:
>
> - VMesh jointly optimizes NeRF and SDF, then extracts meshes/voxels using Marching Cubes and focuses on accelerated rendering with a mesh–voxel structure. After extraction, the mesh–voxel geometry is fixed.
> - Radiance Surfaces optimize occupancy fields and then extract meshes using TSDF or Marching Cubes.
>
> In contrast, our method optimizes meshes end-to-end, avoids reliance on post-optimization meshing algorithms, and achieves accurate reconstructions with compact, well-structured topology (see Figure 1 in the main paper), which is crucial for downstream applications such as physical simulation.

---

> ### Author Response · Authors · 2025-11-15
> **part 2**
>
> ## 3. Experimental Results: Performance on Thin Structures
>
> ### 3.2.1 Thin Structure Performance in the paper
>
> The reviewer requests **"comparisons on the Ship or Ficus scenes from the NeRF Synthetic dataset"** to evaluate reconstruction performance on thin structures. In the main paper, thin-structure performance is demonstrated on Scan 37 (Figure 4) and the Dragon/Monster examples (Figure 5). For Scan 37, our method successfully reconstructs complete scissors, whereas volumetric methods such as Gaussian Surfels and GOF produce broken structures—highlighting the advantage of mesh connectivity. When switching to pure mesh optimization (second stage, without DMTet reparameterization), the scissor tip shows sensitivity to local mesh resolution, in contrast to IMLS-Splatting, which retains a grid-based reparameterization throughout training. This suggests that, for extremely thin structures, maintaining a stable reparameterization during the entire optimization is beneficial. Approaches such as IMLS-Splatting, DMTet, or TetSphere can preserve topology in this way. However, these methods still rely on traditional supervision signals (normals/shading/depth), while our method introduces a more effective photometric supervision scheme. These two directions are complementary and could potentially be combined.
>
>
> ### 3.2.2 Additional Comparision on NeRF Synthetic dataset
>
> We provide additional results on the NeRF Synthetic Dataset to further highlight our method's effectiveness in capturing thin structures, as shown **in the revised manuscript (10th page) uploaded to the OpenReview system**. In Figure 8, we compare our approach with Gaussian Surfel. The results demonstrate that our method accurately represents thin structures and consistently outperforms Gaussian Surfel—particularly for the mast of the ship and the vase in the ficus example. Furthermore, our approach achieves these improvements with significantly reduced vertex counts and optimized mesh topology, as evident in the leaf blade of ficus.
>
> These improvements can be attributed to two main factors. First, our method introduces accurate volumetric supervision to meshes, enabling the model to learn fine geometric details such as the wrinkles on the vase and the mast in the ship example. Second, by employing remeshing during optimization and avoiding post-optimization meshing, our method achieves better mesh topology with fewer vertices, which is critical for downstream applications such as physical simulation.
>
> We also observe that the cable of the ship is not produced by either method. For our approach, this suggests that isotropic remeshing may not be well-suited for extremely thin structures such as cables or other fine details like human hair. In this case, adaptive remeshing could be explored to generate slender triangular faces for cable-like structures while maintaining isotropic faces for flat regions. For Gaussian Surfel, although Gaussian ellipsoids can be positioned to represent cables, it fails to reconstruct meshes from them due to limited meshing resolution, resulting in accumulated errors during ellipsoid-to-mesh conversion.

---

> ### Author Response · Authors · 2025-11-15
> **part 3**
>
> ## 4. Discussion on Dependence on Tetrahedral and Extension to Scene-level Data
>
> The reviewer states that **"The method still relies on tetrahedral grids, which may restrict its scalability and generalization to large-scale or unbounded scenes"** and requests to see **"how to adapt or extend the DMTet initialization to achieve reconstruction at scene scale"** and **"explore FlexiCubes as an alternative to DMTet."** Contrary to this comment, our core contribution is the introduction of volumetric-style photometric supervision for meshes by converting them into a pseudo-volumetric representation. This strategy does not rely on tetrahedral grids and can optimize mesh geometry as long as the deformed layers of the initial mesh coarsely cover the real surface. We validate this claim in Figure 7, where we initialize the mesh with an off-the-shelf method (Gaussian Surfels) and then optimize the geometry using our supervision. The results show that our method recovers accurate geometry from coarse initial meshes.
>
> **Discussion on Mesh Reparameterization**: In our pipeline, DMTet is employed to stabilize topology during the early optimization stage and is a supporting engineering choice. Many works propose diverse reparameterization strategies (e.g., FlexiCubes, IMLS-Splatting, DMTet, TetSphere [5]), which effectively stabilize mesh topology while maintaining representation capability for thin structures. However, these methods still optimize with traditional mesh-based supervision signals such as normals, shading, and depth. Our method takes a complementary approach by introducing fundamental photometric supervision that surpasses existing signals. These two directions are indeed complementary and could potentially be combined. Replacing DMTet with FlexiCubes would be a technical choice that does not affect the main contribution of our introduced mesh-based volumetric supervision.
>
> **Potential for Large-scale Scene-level Reconstruction**: If the reviewer’s comment refers to performance on large scene-level datasets, we acknowledge that our method faces challenges similar to other mesh-based modeling paradigms such as IMLS-Splatting. To extend to large-scale scene reconstruction, such as world or city-scale modeling, additional adaptations would be needed. For example, Street Gaussians [6] could be used to generate an initial mesh, while clipping invisible triangles during rendering would improve efficiency. Additionally, adaptive remeshing should be considered to reduce the number of triangle faces while maintaining quality, assigning more triangles to complex regions and merging small triangles on flat surfaces.
>
> ---
>
> References:
>
> [1] Vmesh: Hybrid volume-mesh representation for efficient view synthesis
>
> [2] Radiance surfaces: Optimizing surface representations with a 5D radiance field loss.
>
> [3] 3D Gaussian Splatting for Real-Time Radiance Field Rendering
>
> [4] Tetrahedron Splatting for 3D Generation
>
> [5] TetSphere Splatting: Representing High-Quality Geometry with Lagrangian Volumetric Meshes
>
> [6] Street Gaussians: Modeling Dynamic Urban Scenes with Gaussian Splatting

---

> > ### Comment · Reviewer_hBem · 2025-11-24
> >
> > Thank you for your response and most of my concerns have been addressed.
> >
> > I suggest including the above discussion in the revised version of the paper. In addition, it would be helpful to include *2DGS* in the qualitative comparison, as it is also a strong baseline. I also have a further concern that the results in Figure 8 indicate that the method does not handle reflections well (e.g., in the *Ship* example). If reflective scenes can be reliably reconstructed, I suggest including additional such cases. If not, I recommend that the authors explicitly acknowledge this limitation in the paper.

---

> > > ### Author Response · Authors · 2025-11-24
> > >
> > > Thank you for your reply and for recognizing our response.
> > >
> > > We will add the discussion above and the qualitative comparison with 2DGS in the revised paper. For 2DGS, it uses a similar optimization approach as Gaussian Surfel. Both methods work with flat Gaussian ellipsoids and face similar problems, especially needing meshing as a post-processing step.
> > >
> > > For the reflective regions in Figure 8, our method can reconstruct detailed water surfaces in the ship example, even with reflections present. If we build a high-quality appearance model that directly considers reflections, the results should be even better. This idea is shown in works like Ref-NeRF [1], which improves NeRF to handle reflections.
> > >
> > > Even without these extra improvements, our method already shows **strong results in mesh reconstruction**—a task that has been difficult for previous methods, even with careful design. We hope our method can be a **pioneering approach**, using meshes for multi-view shape modeling, alongside other volumetric methods like NeRF, SDF, and 3DGS.
> > >
> > > [1] Ref-NeRF: Structured View-Dependent Appearance for Neural Radiance Fields

---

### Official Review · Reviewer_q1sR · 2025-10-26

**Soundness:** 3
**Presentation:** 3
**Contribution:** 3
**Rating:** 4
**Confidence:** 4

**Summary:**

The authors propose a method for reconstructing 3D shapes from multi-view images. They introduce a pseudo-volumetric representation called a soft mesh, assigning per-vertex opacity and features to enable alpha blending. Furthermore, they use DMTet to initialize mesh topology and continuously remesh the shape to improve topological stability. Experiments show that the proposed method achieves high quality shape reconsturction on DTU and BlendedMVS datasets.

**Strengths:**

Strengths
1.The paper shows that optimizing the proposed soft mesh outperforms optimizing a single layer.
2.The reconstruction is both fast and memory-efficient.

**Weaknesses:**

Weeknesses:
1.The performance advantage of the soft mesh is not well justified. While the authors claim that its pseudo-volumetric nature facilitates optimization, an alternative is to convert the base mesh into an SDF and train with NeuS or VolSDF, which remain differentiable with respect to the mesh. The proposed soft mesh seems to approximate this idea primarily for faster rendering. Intuitively, the SDF-based variant should be weaker than standard NeuS/VolSDF because it is constrained by the topology of the base mesh and thus sacrifices flexibility; nevertheless, the reported results indicate the soft mesh surpasses NeuS. Please explain the mechanisms behind this improvement.
2.The procedure for optimizing DMTet in the initial 5,000 iterations is not described. Please elaborate on the optimization objectives, and constraints used during this phase.
3.The explanation of the alpha weights is insufficient. The method adopts VolSDF’s mapping, which was originally used to compute attenuation coefficients; the authors should justify its suitability for alpha weights, additionally clarify whether the expected depth or the most visible point lies on the base mesh.
4.The paper does not report first-stage reconstruction quality. Including quantitative metrics for this stage would clarify the effectiveness of the subsequent stage.

**Questions:**

The author should provide more explanation. I am open to adjusting my rating since representing 3D scenes with multi-surface is critical and might be a potential research direction in the future, although the performance of the proposed method is poor.

---

> ### Author Response · Authors · 2025-11-15
> **part 1**
>
> ## 1. Clarification
>
> ### 1.1 Our Motivation
>
> The reviewer states that **"the pseudo-volumetric nature facilitates optimization; an alternative is to convert the base mesh into an SDF and train with NeuS or VolSDF, which remain differentiable with respect to the mesh."** This "alternative" may refer to initializing an SDF from a mesh and applying a loss to reduce the SDF values at mesh vertices, thereby attempting to align the mesh with the implicit 0-level isosurface of the SDF. However, this approach does not guarantee that the loss will reach zero, nor does it assure perfect alignment between the mesh and the SDF's 0-level isosurface. This misalignment will cause accumulated error when optimizing the SDF but using the mesh as surface reconstruction, sharing a similar problem with post-optimization meshing strategy. In contrast, our method calculates SDF for deformed layers with respect to the base mesh, ensuring the base mesh to align with the 0-level isosurface and providing more accurate surface reconstruction. Moreover, jointly optimizing a mesh and SDF is limited by the low efficiency of SDF-based methods, which are much slower than our direct mesh optimization approach using tile-based rasterization.
>
> In addition, we have considered another possible solution: modeling mesh-based SDFs with differentiable point-to-mesh distance calculations. For example, we could sample points via ray marching, project them onto the mesh surface, and compute the point-to-mesh distance as the signed distance for volumetric rendering and mesh geometry optimization. However, this approach incurs heavy computational cost, as finding the closest point on the mesh for densely sampled points is very expensive and makes the design impractical. In contrast, our method converts the "densely sampled points" to "sparsely sampled layers" by modeling the mesh-based SDF only around the mesh and sampling sparse layers within that region. This greatly improves rendering efficiency and makes mesh-based SDF design practical.
>
> Beyond these two possible solutions, we are not aware of other alternatives. To the best of our knowledge, we are the first to consider mesh-based SDFs and propose a practical pipeline for optimizing them with pseudo-volumetric rendering.
>
>
>
> ### 1.2 Better Performance than NeuS
>
> The reviewer comments that **"the SDF-based variant (the proposed method) should be weaker than standard NeuS/VolSDF because it is constrained by the topology of the base mesh and thus sacrifices flexibility."** We respectfully disagree and assert that our method outperforms NeuS/VolSDF. NeuS/VolSDF employ a coordinate-based MLP and extract meshes via Marching Cubes. However, MLPs are known to struggle with representing high-frequency details (as also noted in the Gaussian Surfels literature), whereas explicit representations such as 3DGS or meshes excel in this area. Directly modeling with these explicit representations provides our method a unique advantage in capturing high-frequency details, as is evident in Tables 1 and 2 and Figures 4 and 5 of the main paper.
>
> Additionally, post-optimization meshing often introduces accumulated errors during the conversion process and makes it difficult to control mesh topology. In contrast, our method directly optimizes an explicit and topologically stable mesh, thereby avoiding these post-processing issues. By employing an iterative optimization-remeshing procedure, our approach eliminates conversion-related error accumulation and guides the optimization gradually toward a higher-quality final result.
>
>
>
> ## 2. Implementation Details
>
> ### 2.1 Optimization with DMTet
>
> The reviewer requests **"elaboration on the optimization objectives and constraints used during this phase (DMTet)."** In this stage, we extract a mesh using DMTet, soften it into multiple layers, render it via mesh splatting, and optimize the rendered image with a photometric rendering loss. For mesh smoothness regularization, we employ standard mesh smoothness losses using the official PyTorch3D implementations, including normal consistency and Laplacian smoothing losses. Additionally, since the tetrahedral grid can be viewed as a discrete signed distance field (SDF), we include an Eikonal regularizer (as in NeuS) to encourage unit-norm gradients. We compute per-tetrahedron gradients following Tetrahedral Splatting and enforce $\Vert \mathbf{g} \Vert_2 \approx 1$ via:
>
> $\mathcal{L}_{eik}=\sum (\Vert \mathbf{g} \Vert_2 -1)^2,$
>
> where $\mathbf{g}$ is the gradient within a tetrahedron. We plan to release our code upon acceptance to facilitate reproducibility.

---

> ### Author Response · Authors · 2025-11-15
> **part 2**
>
> ### 2.2 Explanation of Alpha Weights
>
>
> The reviewer requests **"explanation of the alpha weights"** and raises concerns about **"whether the expected depth or the most visible point lies on the base mesh."** In the VolSDF method, SDF values are mapped to density and further converted to blending weights for volumetric rendering. From the equation below (Please see Equation (3) in our main paper, the equation is not fully complied by OpenReview), outer points with SDF values $s > 0$ have smaller density $\alpha$, while inner points with $s < 0$ have larger density:
>
> $
> \\alpha=\\frac{1}{\\beta} (1-\\frac{1}{2}e^{s/\\beta}), s < 0,
> $
>
> $
> \\alpha=\\frac{1}{2 \\beta} e^{-s/\\beta},  s \\ge 0,
> $
>
> where $s$ indicates SDF values and $\beta>0$ is a learnable controlling parameter. Combined with the volumetric rendering equation (Equation (6) in our main paper), outer points have larger accumulated transmittance $\prod_{k=1}^{i-1} \big(1-\alpha_{k}\big)$, while inner points have smaller transmittance:
>
> $
> \\mathbf{C}_{\\mathbf{p}} = \\sum\_{i \\in \\mathcal{N}} \\mathbf{c}\_{i} \\alpha\_{i} \\prod\_{k=1}^{i-1}\\big(1- \\alpha\_{k}\big).
> $
>
> Combining these equations, VolSDF has shown that sample points with SDF values near zero receive the largest blending weights $\\alpha_{i} \\prod_{k=1}^{i-1} \\big(1 - \\alpha_{k} \big)$ during composition.
>
> In our method, we treat the base mesh as the 0-level isosurface and calculate signed distances for deformed layers. When rendered with Mesh Splatting, sample points for the volumetric rendering equation are identified as intersection points of pixel rays and the multi-layer mesh, with density $\alpha$ converted from signed distance $s$. Therefore, sample points with SDF values near zero will have larger blending weights, indicating that **"the expected depth or the most visible point lies on the base mesh."**
>
>
> ## 3. Experimental Results
>
> ### 3.1 First-stage Reconstruction Quality
>
> The reviewer requests **"reporting first-stage reconstruction quality."** In Figure 6 and Table 4 of the main paper, the "DMTet (128)" entry uses DMTet throughout training and can be viewed as purely first-stage optimization. The resulting geometry is very coarse due to limited tetrahedral resolution. Increasing the resolution ("DMTet (256)") improves accuracy somewhat, but DMTet’s mesh extraction is unevenly tessellated, and resolution increases cubically, causing substantial memory growth with limited accuracy gains. By contrast, our full model switches to direct mesh optimization in the second stage and achieves significantly more accurate reconstruction with similar training time. The supplementary material visualizes the full optimization process: the first stage captures coarse geometry, and the second stage sharpens details.
>
>
> ### 3.2 Addressing the Comments of Poor Performance
>
> We respectfully disagree with the reviewer’s assessment that **"the performance of the proposed method is poor."** As our results demonstrate, our method is competitive and, in many cases, stronger than existing approaches.
>
> ### 3.2.1 Object-centric reconstruction quality.
>
> Our method outperforms most SOTA baselines, particularly on complex geometries such as those in BlendedMVS. For example, on BlendedMVS in Table 2, our method reduces reconstruction error by roughly $30\\%$ relative to the previous SOTA Gaussian Surfels $((2.46 − 1.71)/2.46 \approx 0.30)$. On DTU, our quantitative metrics are on par with SOTA, while Figure 4 shows clear improvements in fine details and mesh completeness (e.g., the eye in Scan 106). Additionally, our results exhibit more uniform mesh tessellation (see Figure 1), which is valuable for downstream applications such as physical simulation.

---

> ### Author Response · Authors · 2025-11-15
> **part 3**
>
> ### 3.2.2 Potential for large-scale scene-level reconstruction.
>
> If the reviewer’s comment refers to performance on large scene-level datasets, we acknowledge that our method faces challenges similar to other mesh-based modeling paradigms such as IMLS-Splatting [1]. As the reviewer noted, scaling mesh-based optimization to large scenes while maintaining accuracy, consistency, and efficiency is an important direction for future work. In Figure 7 of the main paper, we use Gaussian Surfels [2] as an off-the-shelf technique to provide a coarse initial mesh and show that our method can accurately reconstruct scene-level data from this initialization. To further extend to large-scale scene reconstruction, such as world or city-scale modeling, additional adaptations are needed. For example, Street Gaussians [3] could be used to generate an initial mesh, while clipping invisible triangles during rendering would improve efficiency. Additionally, adaptive remeshing should be considered to reduce the number of triangle faces while maintaining quality, assigning more triangles to complex regions and merging small triangles on flat surfaces.
>
> Although further challenges remain, we believe our current method provides valuable insights and introduces a fundamental, efficient, robust, and accurate supervision alternative to existing mesh optimization strategies (such as normal, shading, and depth), which could help more robust large-scale mesh optimization.
>
> ---
>
> References:
>
> [1] IMLS-Splatting: Efficient Mesh Reconstruction from Multi-view Images via Point Representation
>
> [2] High-quality Surface Reconstruction using Gaussian Surfels
>
> [3] Street Gaussians: Modeling Dynamic Urban Scenes with Gaussian Splatting

---

### Official Review · Reviewer_Cj7U · 2025-10-30

**Soundness:** 3
**Presentation:** 2
**Contribution:** 2
**Rating:** 6
**Confidence:** 3

**Summary:**

*Motivation:*
- volumetric methods produce overly dense sampling and can be hard to mesh
- surface methods do not req meshing but make it hard to model intricate geometric details

*Method:*
- authors introduce a "soft mesh" representation that comprises multiple semi-transpatent
mesh layers that allow for underlying base mesh to be optimized
- this mesh representation is optimized via "differentiable Mesh Splatting" - a differentiable
rasterization algorithm that supports transpatent triangles, directly supervised by photometric loss. The final color is obtained by passing a set of interpolated (via barycentrics) features to an MLP and integrated via volumetric rendering.

*Evaluation:*
- Method performs favorably compared to several strong baselines (SuGaR, 2DGS, etc)

**Strengths:**

*Clarity:*
- The method description is clear and experimental evaluation is overall well-documented.

*Originality / significance:*
- Proposed approach is interesting as it can be seen as a hybrid formulation
between volumetric and surface-based representations, and allows for direct
mesh extraction (which is very useful e.g. for physics simulations).

*Evaluation:*
- Method performs favorably with respect to several strong baselines on
surface reconstruction task.
- The mesh splatting method seems more efficient than existing
differentiable rendering approaches (nvidiffrast).

**Weaknesses:**

*Clarity/Motivation:*
- The motivation for the approach is not particularly clear - authors point out to
issues with volumetric rep-s and mesh rep-s, and use the concept of receptive field - which is
not clearly defined. My best guess to what they mean is: meshes are typically optimized per-vertex - and thus the corresponding parameterization has a large number of effective degrees of freedom, and thus. However, there is a number of very established parameterizations (control points, ARAP, spectral methods) that do not share this weakness. Similarly, additional
reguralization terms and multi-node constraints (e.g. local smoothess) effectively act
as a way to restrict the "receptive field" (aka reduce effective degrees of freedom).

*Method Limitations:*
- As indicated by authors themselves (Sec 4.4), the method is severely limited by the
quality of the underlying base mesh - so it would not work well in settings with poor quality
of base meshes obtained by off-the-shelf mesh reconstruction. Because of this, the representation and algorithm introduced in the paper itself can be considered more
of a mesh refindement strategy than a standalone algorithm.
- (minor) 20 minutes per scene does not seem like performance suitable
for some real-world applications.

*Novelty:*
- (minor) The idea of using a soft mesh is not particularly new: e.g. [Esposito'2024] introduces
a very similar representation. And more generally, it can potentially be seen as a variant of control point parameterization or neural cages [Yifan'2020].
- (minor) MeshSDF [Remelli'2020] introduces a differentiable iso-surface extraction method, which allows differentiating through meshing. This sounds like a relevant related work.

*Experimental Evaluation:*
- A reasonable set of baselines is used for volumetric reconstruction, but potentially it
would also be interesting to compare against other differentiable rendering frameworks with "standard" differentiable rendering (nvdiffrast or DRTK) and an off-the-shelf mesh parameterization (e.g. mesh + smoothness).
I do appreciate the performance comparison with nvdiffrast-based iterative rendering implementation, but I wonder if there is a way to dissect
quality wrt to a) mesh representation b) rendering algorithm.
- It is not fully clear why the set of methods is different between Table 1 and Table 2.
- I understand that authors focus specifically on mesh reconstruction, but since the method
actually uses volumetric rendering and produces opacity and color, it seems natural to conduct
evaluation in terms of rendering quality - this might also bring up potential issues on complex
geometries for which meshes are a poor representation. I believe it would be helpful for the readers to understand if the method is also useful as a NVS approach.

**Questions:**

1. Is the "receptive field" a common terminology in the context of geometry representations?
- L034-035: "have a large receptive field along rays" - unclear what is meant here?
- L037: "... only boundary geometry without "
Both remarks are a bit confusing (and both abstract and intro contain the exact same wording which does not make it any more clear in intro) because some volumetric surface representation (e.g. SDF) also in a way only model boundary / surface.
Why relying on priors is a inherently bad thing? And what exactly is meant by "shading or normals"? Does this mean that one has to introduce an image formation model for the underlying
mesh representation to be optimizable? Clarification of this would be appreciated.

2. What is the main bottleneck in the speed/memory wrt to e.g. Gaussian splatting?
Authors did allude that implementation can be improved, but it would be helpful
to understand if there are any fundamental constraints? E.g. since there is a need for sorting triangles - potentially one might have performance issues on object borders/scenes where large
number of triangles fall in the same pixel.

3. Why not have the same set of methods in evaluation on different benchmark datasets (Table 1 vs Table 2)?

---

> ### Author Response · Authors · 2025-11-15
> **part 1**
>
> ## 1. Clarification of Contribution
>
> This section addresses the following **reviewer questions**:
>
> - What exactly is meant by "shading or normals"?
>   - Why is relying on priors inherently a bad thing?
>   - Does this mean that one has to introduce an image formation model for the underlying mesh representation to make it optimizable?
> - The concept of receptive field is not clearly defined
>   - Additional regularization terms and multi-node constraints (e.g., local smoothness) effectively act to restrict the "receptive field" - (i.e., reduce effective degrees of freedom)
> - The motivation for the approach is not particularly clear
>
>
> ### 1.1 Limitation of Existing Mesh-based Surface Reconstruction
>
> Mesh-based surface reconstruction enables end-to-end optimization and avoids the post-optimization meshing typically required by volumetric methods for mesh extraction, thereby reducing accumulated errors and uncontrollable topology resulting from the conversion from volumetric representations to meshes. However, commonly used geometric supervision signals for meshes have notable drawbacks:
>
> - **Normal supervision** requires pretraining a normal estimator to predict normal vector images from color images, using the pixel-wise normal vectors to supervise the mesh normals. However, estimating normals from color images introduces unavoidable estimation errors and dataset bias, especially when input images do not align with the training data distribution.
> - **Shading supervision** relies on a predefined image formation (reflectance) model, such as the Phong model, which typically takes viewing direction, surface normals, light source directions, and surface material parameters as input. These inputs are aggregated into outgoing radiance and compared with the color image. However, a single reflectance model rarely fits all objects and materials, and jointly optimizing geometry, materials, and lighting usually introduces ambiguity.
> - **Depth supervision** uses depth captured by LiDAR or depth sensors, with pixel-wise depth values used to supervise the pixel-wise distance from the mesh to the camera. However, captured depth often contains nontrivial measurement noise.
>
> **Limitation from Reliance on Prior**: Normal estimation depends on large-scale data priors, and shading supervision relies on a predefined reflectance model. Both approaches are brittle when the actual data distribution or materials deviate from these priors. In contrast, we directly optimize using photometric loss with color supervision, which can be easily and accurately captured by cameras and does not require any explicit reflectance model.
>
>
>
> ### 1.2 Concept of Receptive Field
>
> We use the term **receptive field** to describe the spatial region covered by a 3D representation, such that it can receive spatial gradients from sample points inside this field. It is not equivalent to the degrees of freedom of mesh deformation.
>
> - **Volumetric representations** (e.g., SDFs and NeRFs) are typically parameterized by coordinate-based MLPs. Their receptive field is a 3D volume defined by the input coordinates. For instance, Instant-NGP constrains inputs to a bounded domain via a hashed multi-resolution grid, so its receptive field is the region covered by that domain.
> - **Surface representations** (meshes) occupy only a 2D manifold in 3D space. Meshes can only perceive points on their own surface; points outside this 2D manifold are meaningless for meshes.
>
> **Why extend receptive field for meshes?** When optimizing volumetric representations with ray-marching rendering, dense samples are taken along rays and some samples will lie near the actual surface, providing informative gradients for geometry optimization. However, for meshes, if the initial mesh does not overlap the true surface, it cannot perceive the real surface to obtain spatial gradients that move the mesh toward the correct geometry when using only photometric supervision.
>
> Clarifications related to specific manuscript lines:
>
> - **L034–035**: "Volumetric representations have a large receptive field along rays" means we can densely sample points along rays, and some samples will intersect or approach the true surface, providing geometric gradients to optimize the volumetric field.
> - **L037**: "Surface methods avoid meshing but capture only boundary geometry with a single-layer receptive field" means meshes only occupy a 2D layer of the 3D volume. If the current mesh does not coincide with the real surface, it cannot perceive the real surface and generate spatial gradients to move toward it.

---

> ### Author Response · Authors · 2025-11-15
> **part 2**
>
> ### 1.3 Our Contribution
>
> We convert a single mesh into a multi-layer mesh by deforming several layers from a base mesh at randomized offsets. One of these layers is likely to intersect the true surface. This creates a pseudo-volumetric representation that supplies spatial gradients to pull the base mesh toward the real surface. The benefits are:
>
> - **Efficiency**: Tile-based rasterization enables fast optimization.
> - **Robustness**: Supervision comes directly from color images, without reliance on learned priors or fixed reflectance models.
> - **Accuracy**: The enlarged receptive field increases the probability of receiving informative gradients.
>
>
> ## 2. Comparison with More Methods
>
> The reviewer states **"The idea of using a soft mesh is not particularly new"**, mentioning Volumetric Surface [1], Neural Cages [2], and MeshSDF [3]. However, these methods differ significantly from ours and perform far worse on mesh reconstruction tasks. We elaborate on these comparisons below.
>
> ### 2.1 Volumetric Surface
>
> Volumetric Surface extracts a base mesh from volumetric reconstructions and samples multiple layers around it to accelerate volumetric rendering by quickly locating ray–surface intersections. However, these sampled layers are not differentiable with respect to the base mesh and remain fixed after initialization. As a result, the reconstructed mesh is simply the output of the volumetric method, carrying its errors and topology issues.
>
> By contrast, our layers are differentiable with respect to the base mesh, so rendering losses back-propagate and update the base mesh efficiently, robustly, and accurately.
>
>
> ### 2.2 Neural Cage
>
> Neural Cage wraps a mesh with a sparse "cage" and binds the mesh to the cage to regularize mesh deformations. This addresses a different goal—regularizing mesh deformations—rather than recovering accurate geometry from images. In comparison, Neural Cage requires accurate geometry as input, whereas our method recovers accurate geometry from multi-view images.
>
>
> ### 2.2 MeshSDF
>
> MeshSDF introduces differentiable Marching Cubes, which can extract a mesh from an SDF and propagate gradients from the mesh to the SDF. However, MeshSDF still relies on normals, shading, or depth to supervise the mesh and obtain gradients. Our method instead uses photometric supervision on meshes, which is more efficient, robust, and accurate than existing mesh supervision. While one could apply similar supervision to a volumetric field and then extract a mesh with differentiable meshing after optimization, if the mesh is not used during optimization, differentiable meshing offers no benefit over standard meshing algorithms.
>
>
> ## 3. Experimental Results
>
> This section addresses the following **questions issued by the reviewer**:
>
> - Compare against other differentiable rendering frameworks using "standard" differentiable rendering (nvdiffrast or DRTK) and an off-the-shelf mesh parameterization (e.g., mesh + smoothness)
> - Why are the compared methods different between Table 1 and Table 2?
> - Whether the method is also useful as a novel view synthesis (NVS) approach
>
>
> ### 3.1 Optimization with nvdiffrast (IMR) vs. Our Mesh Splatting
>
> As shown in Table 3, our multi-layer mesh can also be optimized using iterative mesh rasterization (IMR) via nvdiffrast, but at a high memory and compute cost. The underlying image formation in both cases is similar: intersect pixel rays with the multi-layer mesh, sort intersections, and composite colors. With unlimited resources, IMR can approach the results of our Mesh Splatting. Our contribution is an efficient, tile-based rasterization and compositing scheme that enables higher parallelism and better memory/computation usage, making training fast and practical.
>
>
> ### 3.2 Initialization with DMTet vs. Mesh + Smoothness
>
> In Figure 6 and Table 4, the "w/o DMTet" ablation corresponds to directly optimizing a mesh initialized as a sphere with smoothness regularization. Compared to the full model, this results in early-stage mesh defects that are difficult to repair, consistent with prior works that reparameterize meshes as volumetric fields (e.g., NvdiffRec, IMLS-Splatting). Using DMTet in the early stage avoids these defects.
>
>
> ### 3.3 Compared Mehtods for Table 1 and 2
>
> Most existing methods target DTU and provide evaluation code, allowing a broader comparison in Table 1. Public evaluation pipelines for BlendedMVS (BMVS) are less common, though BMVS is valuable for its more complex geometry (whereas DTU emphasizes challenging lighting). To ensure meaningful comparison on BMVS, we adapted and trained representative baselines ourselves. The set remains diverse and includes mainstream and strong methods across categories: NeuS (implicit), Gaussian Surfels (explicit), SuGaR and IMLS-Splatting (mesh-based).

---

> ### Author Response · Authors · 2025-11-15
> **part 3**
>
> ### 3.4 Performance on Novel View Synthesis (NVS)
>
> NVS evaluates rendering quality, not geometry fidelity. A fair cross-representation comparison would convert all methods to meshes (e.g., extract a mesh from NeuS via Marching Cubes or from 3D Gaussians via TSDF fusion), then train texture maps on the mesh and render images. This is cumbersome in practice, but in general, more accurate geometry tends to yield better renderings.
>
> Directly comparing mesh-based methods to volumetric ones is less fair: the mesh representation, chosen for compatibility with graphics pipelines and downstream applications, can under-represent extremely intricate or fuzzy structures (e.g., hair), which are easier to model with volumetric or Gaussian primitives. A hybrid approach that combines meshes for surfaces with Gaussian ellipsoids for very fine details is a promising direction for future work.
>
>
> ## 4. Discussion of Limitation
>
> ### 4.1 Dependence on an Initial Mesh
>
> The reviewer comments that **"the method is severely limited by the quality of the underlying base mesh"**, **"it would not work well in settings with poor quality base meshes obtained by off-the-shelf mesh reconstruction"**, and the proposed method is **"a mesh refinement strategy"**. However, our method does not require an accurate initial mesh; it only needs the deformed layers to roughly cover the true surface. We have already validated this in Figure 7 for scene-level datasets, where we initialize with an off-the-shelf coarse reconstructor (e.g., Gaussian Surfels). The results show that our method can recover accurate geometry from poor-quality meshes produced by off-the-shelf mesh reconstruction.
>
> We regard our method as a fundamental optimization strategy for meshes—an efficient, robust, and accurate alternative to the prevalent normal, shading, and depth supervision. It is not merely a refinement technique that assumes a high-quality initialization and only enhances details; our approach can start from a sphere (see the supplementary video) and still converge to accurate geometry. Moreover, the requirement for an initial mesh for scene-level datasets is also applicable to other mesh-based pipelines (e.g., IMLS-Splatting [4]) and, in our view, is not a critical limitation.
>
>
> ### 4.2 20-minute Optimization
>
> The reviewer notes that **"20 minutes per scene is not suitable for some real-world applications"**, asks about **"the main bottleneck in speed/memory compared to, e.g., Gaussian Splatting"**, and observes **"sorting triangles may be slow when a large number of triangles fall in the same pixel"**. Our Mesh Splatting is similar to Gaussian Splatting (GS), which splats primitives into image space, sorts overlapping contributions per pixel, and composites colors. The main difference is that we splat triangles instead of Gaussian ellipsoids. As such, increasing the number of primitives to be sorted is a common concern for both our Mesh Splatting and GS. This can be further optimized with techniques such as terminating projection when accumulated density exceeds a threshold. Further speedups are possible by borrowing design ideas from GS—such as adaptive remeshing to reduce triangle counts, analogous to density control in GS. However, these are engineering improvements and do not affect the core contribution: converting a mesh into a pseudo-volumetric representation to enable volumetric-style photometric supervision for end-to-end surface reconstruction. Furthermore, our training is already substantially faster than ray-marching methods such as NeuS (~20 minutes vs. >200 minutes in our settings).
>
>
> ---
>
> References:
>
> [1] Volumetric Surfaces: Representing Fuzzy Geometries with Layered Meshes
>
> [2] Neural Cages for Detail-Preserving 3D Deformations
>
> [3] MeshSDF: Differentiable Iso-Surface Extraction
>
> [4] IMLS-Splatting: Efficient Mesh Reconstruction from Multi-view Images via Point Representation

---

### Author Response · Authors · 2025-11-15

Thanks for the thoughtful comments and for recognizing the value of our approach. We carefully reviewed all points and provide our responses. We are happy to further elaborate if needed.

---

### Author Response · Authors · 2025-11-30
**Summary of reviews for new AC**

We regret that the review and rebuttal process was affected by the data leakage bug. We sincerely thank the new Area Chair for checking the reviews and our response, and for continuing with the discussion period. To reduce the Area Chair’s workload, we provide a brief summary of the reviewers’ comments and our responses below.


## 1. Summary of reviewers’ comments

**All reviewers agree that our work is novel, well-motivated, and makes a strong contribution**. Most concerns are about needing more explanation of our method, more comparison with related methods, and how our approach adapts to large-scale, scene-level datasets. We believe our response has addressed these points. Before the data leakage event, **reviewer hBem** said our reply had solved most of his concerns. The other two reviewers have not replied yet, but we think our response covers their questions as well. Only **reviewer q1sR** gave us a low score (4), but he also said, **"I am open to adjusting my rating"** if we explain things more, which we have done.

The main concerns from **reviewer q1sR** are:

- Why our method does better than NeuS
- How we optimize with DMTet
- How the alpha weights work
- The quality of the first-stage reconstruction

In our reply to **reviewer q1sR**, we addressed each of these issues in detail. We also noticed that **reviewer q1sR** may have ***misinterpreted the performance of our method***, commenting the performance "was poor". However in fact, ***our method outperforms most SOTA baselines*** as stated in our paper. For example, in Table 2, our method reduces the reconstruction error (Chamfer Distance in *cm*) by about $30\\%$ compared to previous SOTA Gaussian Surfels.

If the "poor performance" comment is about large scene-level datasets, we agree that our method faces ***similar challenges as other mesh-based methods***. We have proposed ways to adapt our method for large-scale scene reconstruction in our response to **reviewer q1sR**. Also, Figure 7 in our paper shows that ***our method can accurately reconstruct scene-level surfaces*** when initialized with a coarse mesh from Gaussian Surfels.

With these responses, we trust that **reviewer q1sR** will reconsider his score. We hope this will be considered in the final decision.


## 2. Brief introduction to our method

Our method introduces a new mesh optimization strategy by softening the mesh into several differentiable layers. Each layer is a sample of Signed Distance Fields (SDFs) around the base mesh. We use splatting-based rasterization to project these layers into image space. The overlapping projections are then blended using weights based on each layer’s distance to the base mesh. During optimization, we compare the rendered results with the real images, which generates gradients that help move the base mesh toward the true object surface. We also use mesh reparametrization (DMTet) and remeshing techniques to balance mesh topology stability and optimization efficiency.

Our approach brings meshes to SOTA surface reconstruction, a field currently dominated by volumetric methods like 3DGS, NeRF, and SDF. Compared to these methods, which always require meshing as a post-processing step, our method allows for more efficient, accurate, and topology-controllable mesh reconstruction. **Before our method, mesh optimization usually relied on shading, normal, or depth supervision, each having its own limitations compared to the multi-view photometric supervision we use**.

---

### Meta-Review · Area_Chair_39Vz · 2026-01-05

**Summary:**

This paper presents a method for 3D mesh optimization where the relaxation is inspired by volume rendering.
The submission received mixed reviews from the reviewers.
The reviewers mainly recognize the innovation of the approach, strong empirical results, improved efficiency in performance, and clear presentation.
The main concerns from the reviewers were the dependency on sufficiently good initial meshes (Cj7U and hBem), originality of the idea (Cj7U and hBem), and gaps in evaluation (all).
After reading the paper, the reviewers' comments and the authors' rebuttal, the AC believes the authors' responses would have addressed the reviewers' major concerns, and the merits from the paper outweighs the weaknesses. The AC thus recommends acceptance.

**Reviewer Concerns:**

Reviewers' concerns mostly addressed:
- Concept of "Receptive Field" (Cj7U)
- Motivation for soft meshes vs. depth/normal priors (Cj7U and hBem)
- Novelty vs. prior methods (Cj7U and hBem)
- Performance on thin structures (Cj7U and hBem)
- Efficiency bottlenecks (Cj7U)
- Justification on advantage of soft mesh (q1sR)
- Alpha weight logic (q1sR)
- First-stage quality metrics (q1sR)

Outstanding concerns:
- Novel view synthesis evaluation (Cj7U)
- Scalability to large scenes (hBem)
- Initialization with other methods e.g. Flexicubes (hBem)

**Reviewer Scores:**

I think Cj7U and hBem would keep their ratings of 6, and q1sR would raise their rating to 6.

---

### Decision · Program_Chairs · 2026-01-26

Accept (Poster)